# Natural Alkaloids as Multi-Target Compounds towards Factors Implicated in Alzheimer’s Disease

**DOI:** 10.3390/ijms24054399

**Published:** 2023-02-23

**Authors:** Rudolf Vrabec, Gerald Blunden, Lucie Cahlíková

**Affiliations:** 1Secondary Metabolites of Plants as Potential Drugs Research Group, Department of Pharmacognosy and Pharmaceutical Botany, Faculty of Pharmacy, Charles University, Heyrovského 1203, 500 05 Hradec Králové, Czech Republic; 2School of Pharmacy and Biomedical Sciences, University of Portsmouth, Portsmouth PO1 2DT, UK

**Keywords:** Alzheimer’s disease, plant alkaloids, marine alkaloids, multi-target compounds

## Abstract

Alzheimer’s disease (AD) is the most common cause of dementia in elderly people; currently, there is no efficient treatment. Considering the increase in life expectancy worldwide AD rates are predicted to increase enormously, and thus the search for new AD drugs is urgently needed. A great amount of experimental and clinical evidence indicated that AD is a complex disorder characterized by widespread neurodegeneration of the CNS, with major involvement of the cholinergic system, causing progressive cognitive decline and dementia. The current treatment, based on the cholinergic hypothesis, is only symptomatic and mainly involves the restoration of acetylcholine (ACh) levels through the inhibition of acetylcholinesterase (AChE). Since the introduction of the Amaryllidaceae alkaloid galanthamine as an antidementia drug in 2001, alkaloids have been one of the most attractive groups for searching for new AD drugs. The present review aims to comprehensively summarize alkaloids of various origins as multi-target compounds for AD. From this point of view, the most promising compounds seem to be the β-carboline alkaloid harmine and several isoquinoline alkaloids since they can simultaneously inhibit several key enzymes of AD’s pathophysiology. However, this topic remains open for further research on detailed mechanisms of action and the synthesis of potentially better semi-synthetic analogues.

## 1. Introduction

Alzheimer’s disease (AD) is a neurodegenerative disease that progressively worsens with time and seriously affects the daily life and health of elderly people over 65 years old and is characterized by memory loss and behavior abnormalities [1]. *The World Alzheimer Report 2019* pointed out that with the continuous acceleration of the aging population, the number of AD patients worldwide had exceeded 50 million, but the number of patients is expected to double by 2030 and to reach 152 million by 2050. Moreover, AD has a significant influence on health and economic development that needs to be addressed urgently [2].

AD, the most common cause of dementia, is caused by a combination of genetic factors, endogenous factors, exogenous environment, and many other risk factors. The vast majority of AD cases arise sporadically, either in the form of late-onset AD in individuals ≥ 65 years old (~90% of patients) or as early-onset AD when patients become AD symptomatic between ~45–65 years old (~6–16% of patients). A small group of patients (~1%) have early-onset autosomal dominant AD caused by mutations in either the PSEN1, PSEN2, or APP genes [3,4]. All these factors cause complex brain changes, which lead to cell damage [5].

Microscopic changes in the brain begin long before the first signs of memory loss. The exact genesis of the disease is not yet fully explained and is constantly under debate and review. To date, several factors have been demonstrated to be responsible for AD development and progression, thus playing an eminent role in the pathogenesis of AD [6]. Moreover, several hypotheses, including cholinergic, amyloid beta (Aβ), hyperphosphorylation of τ-protein, calcium dyshomeostasis, and oxidative stress, have been proposed to explain the pathophysiology of the disease [7,8,9]. The pathological hallmarks of this disorder are extracellular accumulation of Aβ plaques composed of Aβ peptides, neurofibrillary tangles (NFTs) composed of hyperphosphorylated tau protein, brain inflammation, and atrophy. These changes are thought to begin years before the clinical symptoms are noticeable [10,11].

The oldest hypothesis, on which current therapy is still based, is the cholinergic hypothesis that attributes failure in neurotransmission. Cholinergic neuron damage was considered to be a critical pathological change that correlated with cognitive impairment in AD.

Acetylcholine (ACh) is an important neurotransmitter used by cholinergic neurons, which has been involved in critical physiological processes such as attention, learning, memory, stress response, wakefulness and sleep, and sensory information [11,12].

Acetylcholinesterase (AChE, EC 3.1.1.7) is a serine protease that plays a key role in the cholinergic transmission of the central nervous system (CNS) and neuromuscular junctions. The physiological function of AChE is the hydrolysis of ACh to choline and acetic acid [13]. AChE is primarily expressed by neural tissue, neuromuscular junctions, plasma, and erythrocytes [14,15]. There are two structurally different isoforms in the human brain-monomeric G1 and significantly larger amounts of tetrameric G4. In the other tissues, dimeric isoform G2 is also present. AD is characterized by a substantially elevated level of G1 isoform [9]. 

Another cholinesterase (ChE) capable of hydrolysis of ACh is butyrylcholinesterase (BuChE, EC 3.1.1.8). Although BuChE is physiologically accountable only for 20% of ACh’s hydrolysis, in the later stage of AD its activity is increased up to 90% and, therefore, takes over the role in the degradation of ACh [16]. Different isoforms of AChE in the brain and cerebrospinal fluid in patients with AD are connected with abnormal glycosylation [17]. Although the role of AChE in neurodegenerative diseases is well studied, the role of BuChE still remains a bit unclear. This pseudocholinesterase lacks its natural substrate [9]. Some studies have also shown that BuChE can indirectly contribute to the pathophysiology of type 2 diabetes by increasing insulin resistance [18]. The advantage of selective inhibition of AChE/BuChE is still debated, although some authors imply that better selectivity to AChE can lead to fewer side effects [19].

When comparing results from biological in vitro tests focused on the inhibition of ChEs, it is important to know the source of the used enzyme. The common source for in vitro tests with AChE is an eel, *Electrophorus electricus* (*Ee*AChE), which, however, can provide divergent results in comparison to human AChE (*h*AChE) [7,20]. Due to the several dissimilarities in the binding site of AChE from different organisms, the usage of *h*AChE represents a more accurate model for in vitro and in silico studies [21]. Nowadays, recombinant technology for obtaining *h*AChE is much more beneficial than *h*AChE isolated from human serum, as in the past, because its production can be better controlled and standardized [22]. For biological in vitro tests with BuChE, horse (*equine*) serum (*eq*BuChE), human plasma or human recombinant enzyme (*h*BuChE) is usually used. However, BuChE from various sources can, yet again, provide different results [23]. Recently, interest in ChE inhibitors (ChEIs) has increased due to the findings supporting cholinesterase’s involvement in Aβ peptide fibril formation during AD pathogenesis [24]. Various studies have supported that ChEIs can prevent Aβ oligomerization, thus displaying antiamyloid and neuroprotective disease-modifying effects [25].

Glutamate is the most important excitatory neurotransmitter in the human brain, and the regulation of its homeostasis is of utmost importance for the brain’s proper functioning [26]. Excitatory glutamatergic neurotransmission via the *N*-methyl-D-aspartate receptor (NMDAR) is critical for synaptic plasticity and the survival of neurons [27]. NMDA receptors are permeable for Na^+^, K^+^, and highly permeable for Ca^2+^, which acts as a secondary messenger to stimulate intracellular signaling cascades [28]. Activation of NMDR in different ways can lead to either long-term potentiation or long-term depression of synaptic strength. However, excessive NMDAR activity due to increased Ca^2+^ entry into the cell causes excitotoxicity and promotes cell death, underlying a potential mechanism of neurodegeneration that occurs in AD. The glutamatergic hypothesis postulates that the progressive cognitive decline seen in AD patients is due to neuronal cell death caused by the overactivation of NMDA receptors and the subsequent pathological increase in intracellular calcium.

The amyloid hypothesis states that Aβ forms long insoluble amyloid fibrils, which accumulate in senile plaques in critical regions of the brain and are toxic to neurons. The pathogenic Aβ peptides originate from the proteolytic activity on amyloid precursor protein (APP), a naturally occurring transmembrane polypeptide containing 37 to 49 amino acid residues [29]. The precise physiological function of APP is not known, and remains one of the vexing issues in the field. APP undergoes proteolytic cleavage by either a nonamyloidogenic or amyloidogenic pathway. In the former, APP is cleaved by the enzyme γ-secretase, releasing a soluble N-terminal fragment (sAPPα) and a C-terminal fragment (C83), which is further cleaved by α-secretase, releasing a C-terminal fragment of 3 KDa (C3). Within the amyloidogenic pathway, APP is cleaved by β-secretase (BACE-1, EC 3.4.23.46), releasing a smaller N-terminal fragment (sAPPβ) and a C-terminal fragment (C99) that produce the full-length β-amyloid peptides upon the subsequent cleavage by γ-secretase. The formed amyloid peptides consist of 38–43 amino acids, whereas those with 40 and 42 amino acids (Aβ_40_ and Aβ_42_) are the most abundant in the brain [30]. Aβ species are released into various types of assemblies, including oligomers, protofibrils, and amyloid fibrils. Amyloid fibrils are larger and insoluble, and they can further assemble into amyloid plaques, while amyloid oligomers are soluble and may spread throughout the brain [10]. Despite their similarities, Aβ_42_ is more prone to aggregation and fibrilization, being the most toxic Aβ peptide and playing a pivotal role in the pathogenesis of AD [31]. 

As mentioned above, the next crucial neuropathological hallmark of AD is the presence of NFTs of τ protein, which is a key microtubule-associated protein that in healthy neurons binds and stabilizes microtubules by reversible, enzymatically mediated phosphorylation and dephosphorylation processes [32]. When the dephosphorylation is not sufficient, it does not bind adequately to other microtubules and polymerizes into filaments that further form NFTs [33]. Recent evidence points to additional functions for tau. For example, tau phosphorylation enables neurons to escape acute apoptotic death through stabilizing β-catenin [34]. Moreover, tau exerts an essential role in the balance of microtubule-dependent axonal transport of organelles and biomolecules by modulating the anterograde transport by kinesin and the dynein-driven retrograde transport. In the healthy brain, 2–3 residues of tau are phosphorylated. In AD and other tauopathies, however, the phosphorylation level of tau is significantly higher, with approximately nine phosphates per molecule [35]. The common process that plays a vital role in the intensity of tau modification is phosphorylation and dephosphorylation, influenced by specific protein kinases like glycogen synthase kinase-3β (GSK-3β, EC 2.7.11.26), cyclin-dependent kinase 5 (CDK5, EC 2.7.11.22), and C-Jun amino-terminal kinase (JNK, EC 2.7.11.24), which fall under the family of proline-directed protein kinases [36].

GSK-3β is a ubiquitous serine (Ser)/threonine (Thr) protein kinase involved in the transfer of a phosphate group from adenosine triphosphate (ATP) to Ser and Thr acid residues of target substrates. There are two isoforms of GSK-3, GSK-3α, and GSK-3β encoded by two different genes [37]. In the CNS, GSK-3β is the most abundant, and its expression levels are known to increase with age [38]. The predominant hypothesis in AD suggests that GSK-3β is affected by amyloid peptides [39]. In AD, the overactivity and/or overexpression of GSK-3β accounts for memory impairment, tau hyperphosphorylation, increased β-amyloid production, and local plaque-associated microglial-mediated inflammatory responses, which are hallmarks of the disease [40].

CDK5, a proline-directed Ser–Thr protein kinase, plays an important part in the physiological development of the central nervous system, for phosphorylating many relevant substrates [41]. Since the phosphorylation of τ proteins is primarily dependent on GSK-3β and CDK5 [42], inhibition of GSK-3β and CDK5 is accepted as a promising strategy for the treatment of AD [43].

In general, tau-targeting therapies remain challenging because of an incomplete understanding of AD, the lack of robust and sensitive biomarkers for diagnosis and response monitoring, and the obstruction of the blood–brain barrier (BBB) [11].

C-Jun N-terminal kinases (JNKs) are a family of protein kinases that play a central role in stress signaling pathways implicated in gene expression, neuronal plasticity, regeneration, cell death, and regulation of cellular senescence. There are three different human JNKs. Unlike JNK1 and JNK2, JNK3 is mostly expressed in the brain, and only a small portion is expressed in the heart and testis [44]. JNK3 has been considered a potential therapeutic target for neurodegenerative diseases associated with neuronal cell death. It has been shown that there is a JNK pathway activation after exposure to different stressing factors, including cytokines, growth factors, oxidative stress, unfolded protein response signals, and Aβ peptides. In addition, activation of JNK has been identified as a key element responsible for the regulation of apoptosis signals, and, therefore, it is critical for pathological cell death associated with neurodegenerative diseases and, among them, with AD [45]. 

The altered brain levels of monoamine neurotransmitters due to monoamine oxidase (MAO) are directly associated with various neuropsychiatric conditions like AD [46]. Activated MAO induces Aβ deposition via abnormal cleavage of the APP. Additionally, activated MAO contributes to the generation of neurofibrillary tangles and cognitive impairment due to neuronal loss. MAO exists in two forms (monoamine oxidase-A (MAO-A) and monoamine oxidase-B (MAO-B)), and recent neuroimaging studies have shown that the increased MAO-B expression in the brain and platelets in AD starts several years before the onset of the disease [47,48]. However, the mechanism by which MAO-B affects AD pathogenesis is not known. MAO inhibitors have neuroprotective effects related to oxidative stress, which are desirable properties for the development of multi-target drugs for AD.

Current therapy for AD is built around cholinergic and glutamatergic hypotheses. Three AChE inhibitors, namely, donepezil, galanthamine, and rivastigmine, and one fixed combination of donepezil and memantine (approved in 2014) are currently used as the main therapeutic option for AD treatment (Figure 1) [49,50]. However, such therapeutic approaches provide only symptomatic relief for several months. These available drugs are marketed for mild to severe stages of AD.

In June 2021, a new AD drug named aducanumab, sold under the brand name Aduhelm™, was approved by the FDA in the USA for people with mild symptoms of AD, such as individuals who are still independent in basic daily functioning [51]. Aducanumab is a human IgG1 monoclonal antibody, which should be able to reduce brain Aβ deposits. It is the first drug with a putative disease-modifying mechanism for the treatment of this devastating disorder [52]. The decision was highly controversial and led to the resignation of three FDA advisers because of the absence of evidence that the drug was effective, as clinical trials gave conflicting results on its effectiveness [53]. Very recently (January 2023), another monoclonal antibody, lecanemab (Lequembi™), was approved by the FDA’s accelerated pathway [54]. Like aducanumab, this drug is able to clear Aβ plaques and slow the decline of cognition in patients with early-stage AD. However, the treatment is associated with adverse effects, and further studies on its safety and efficacy are still very much needed [55].

AD is a complex disorder involving multiple factors that necessitate the need to identify and develop hybrid molecules that can target two or more pathological changes in the brain of AD patients [56]. Natural products are a rich and interesting source of various structural scaffolds, which can be used for the development of new multi-target compounds towards factors implicated in AD [57,58,59,60]. 

One of the most attractive groups of natural products is, without a doubt, alkaloids, which are produced by a large variety of organisms, including bacteria, fungi, plants, marine organisms, and animals [61]. Alkaloids are a particular group of low-molecular-weight, nitrogen-containing compounds, mainly biosynthetically derived from amino acids resulting in a variety of chemical structures [62]. These natural products demonstrate a wide range of biological activities. Many of them are already used in the therapy of various diseases, like opiate alkaloids, which humans have used for millennia to reduce pain; vinca alkaloids, extracted from *Catharanthus roseus* (L.) G.Don (Apocynaceae); and taxanes, isolated from the genus *Taxus* L. (Taxaceae), which are used as chemotherapy agents [63,64,65]. A further example is the previously mentioned galanthamine, which was originally isolated from *Galanthus woronowii* Losinsk. (Amaryllidaceae) [7] and many other species. 

The present review is a continuation of our previous one, which aimed to summarize the interesting single biological activities of isoquinoline alkaloids [7]. The current review aims to comprehensively summarize the research that has been published on selected natural alkaloids as potential multi-target compounds towards factors implicated in AD. 

## 2. Plant Alkaloids as Multi-Target Compounds for the Treatment of AD

### 2.1. Indole Alkaloids: Ajmalicine and Reserpine

Ajmalicine (AJM) and reserpine (RES) are indole alkaloids (Figure 2) isolated from *Rauvolfia serpentina* Benth. ex Kurz (Apocynaceae), which has been used in folk medicine in India for centuries to treat a wide variety of maladies, including snake and insect bites, febrile conditions, malaria, abdominal pain, and dysentery. It was also used as a uterine stimulant, febrifuge, and cure for insanity. The plant was mentioned in Indian manuscripts as long ago as 1000 BC and is also known as *sarpagandha* and *chandrika* [66].

RES and AJM are used as antihypertensive agents to control high blood pressure [67], but further biological activities have also been studied. Both alkaloids have been studied in vitro for their inhibition potencies of AChE/BuChE, BACE-1, and MAO-B (Table 1), and the obtained results were subjected to in silico analysis [68]. The in vitro neuroprotective potential of the two alkaloids against Aβ toxicity and anti-oxidative stress were studied using PC12 cell culture. RES has been identified as a dual cholinesterase inhibitor with IC_50_ values of 1.7 µM for AChE and 2.8 µM for BuChE, which are, in the case of AChE inhibition, comparable with that of galanthamine. As for BuChE, RES shows up to 15 times higher inhibitory potential compared to galanthamine. The anti-aggregation inhibition potential of RES and AJM was studied through thioflavin T (ThT) fluorescence assay with an evaluation of the red shift in the Congo red (CR) dye binding assay. The ThT probe gives a bright fluorescence at 480 nm upon binding exclusively to Aβ_42_ proto-fibrils. RES and AJM significantly inhibited the formation of Aβ_42_ fibrils in a concentration-dependent manner (11–44 µM), with 69% (at 44 µM) inhibition of fluorescence for RES and 57% (at 44 µM) for AJM. Both alkaloids were also evaluated for their inhibition potential of β-sheet formation of Aβ_42_, which occurs during the oligomerization process of Aβ_42_. This change in the secondary structure was measured by circular dichroism (CD) in the far UV region (200–250 nm). CD spectra of Aβ_42_ co-incubated with RES and AJM showed strong inhibition of β-sheet formation by 64% for RES and 53% for AJM. Both alkaloids also protect PC12 cells (rat pheochromocytoma cells, which are the most commonly used in neuroscience research, including studies on neurotoxicity) against Aβ_42_ (92% for RES at 40 µM; 67% for AJM at 40 µM) and H_2_O_2_ (93% for RES at 40 µM; 89% for AJM at 40 µM) induced cytotoxicity [68,69]. RES and AJM were also screened for inhibition potential of other important targets for AD, the BACE-1 enzyme, and MAO-B. AJM showed the maximum inhibition of BACE-1 activity at 69% (at 50 µM), whereas RES inhibited BACE-1 to 47% at the same concentration. RES and AJM significantly inhibited the MAO-B enzyme at a concentration 10 µM. Both tested compounds showed comparable inhibition potency (82%, 83%, respectively). Molecular docking analysis revealed strong binding of both compounds to the catalytic site of AD targets. 

RES has also been studied for its protective ability against Aβ toxicity in the AD model of *Caenorhabditis elegans*, which is manifested as paralysis [90]. This model expresses the human toxic Aβ_1–42_ in the muscle and helps in understanding the possible mechanism of AD pathology [91]. RES was able to alleviate the AD pathogenesis in *C. elegans* by delaying toxic Aβ expression-mediated paralysis. In addition, RES increased the stress tolerance and extended the lifespan of *C. elegans* at a tested concentration of 60 µM [90].

The ADMET profile of AJM showed a promising profile as a drug candidate with BBB permeability. RES failed to obey Lipinski’s rule of five because of its molecular weight (608.688 g/mol). On the other hand, it showed good potential to cross the BBB. Thus, both indole alkaloids can be recognized as promising multi-target compounds for AD.

### 2.2. β-Carboline Alkaloids: Harmine and Harmaline

The β-carboline alkaloids harmine (HAR) and harmaline (HAL) are the main components (Figure 3) of *Peganum harmala* L. (Nitraceae), the extract of which has showed antimicrobial, antitumor, and antidiabetic potencies [70,92]. HAR and HAL are known as natural MAO inhibitors and are one of the ingredients in the drink ayahuasca, which has been used by native people in South America for centuries for its anxiolytic and antidepressant effects [93]. Both alkaloids possess a variety of biological activities in connection with the potential treatment of AD, including AChE inhibitory, antioxidant, MAO-A, and anti-inflammatory (Table 1), and is able to cross the BBB, which has been evaluated using PAMPA-BBB assay (*P_e_* = 44.6 × 10^−6^ cm/s; CNS+) [71]. He et al. studied the behavioral effects of HAR on scopolamine-induced cognitive impaired mice and APP/PS1 transgenic mice, models for AD, using the Morris Water Maze (MWM) test [94]. Results showed that HAR (20 mg/kg) administered by oral gavage for two weeks could effectively enhance the spatial cognition of C57BL/6 mice impaired by intraperitoneal injection of scopolamine (1 mg/kg) [94]. Moreover, long-term consumption of HAR (20 mg/kg) for ten weeks also slightly benefited the impaired memory of APP/PS1 mice. Furthermore, HAR could pass through the BBB; it has been proposed that HAR might form hydrogen bonds and π–π interactions with the active residues in acetyltransferase [94]. A molecular docking study disclosed that HAR could directly dock into the active catalytic site of AChE.

Li et al. studied the beneficial effect of HAR and HAL on scopolamine-induced cognitive dysfunction in a C57BL/6 mouse model using the MWM test [95]. To elucidate further the potential mechanisms of HAL and HAR in improving the memory of mice after scopolamine administration, the levels of multifarious biochemical factors and protein expressions associated with the cholinergic system, oxidative stress, and inflammation were investigated. The results showed that HAL and HAR could effectively ameliorate memory deficits in scopolamine-induced mice. Both of them exhibited an enhancement in cholinergic function by: inhibiting AChE and inducing choline acetyltransferase activities; antioxidant defense via increasing the antioxidant enzyme activities of superoxide dismutase and glutathione peroxidase; reducing maleic dialdehyde production; and anti-inflammatory effects through suppressing myeloperoxidase, tumor necrosis factor α (TNF-α), and nitric oxide (NO); as well as modulation of critical neurotransmitters such as ACh, choline, L-tryptophan (L-Trp), 5-hydroxytryptamine (5-HT), γ-aminobutyric acid (γ-GABA), and L-glutamic acid (L-Glu) [95]. 

Taken together, both alkaloids could effectively ameliorate memory impairments in a scopolamine-induced mouse model via improvement in cholinergic system function, suppression of oxidative stress and inflammation damage, and modulation of vital neurotransmitters. 

Since GSK-3β and dual-specificity tyrosine phosphorylation-regulated kinase 1A (DYRK1A) have gained a lot of attention for their role in tau pathology, HAR and its semisynthetic derivatives were evaluated as dual GSK-3β/DYRK1A inhibitors. HAR itself showed moderate GSK-3β inhibition potency with an IC_50_ = 32.1 ± 1.0 µM but was a strong inhibitor of DYRK1A (IC_50_ = 0.080 ± 0.007 µM) [71]. Interestingly, GSK-3β and DYRK1A are homologous protein kinases, but HAR has a different binding mode with GSK-3β and DYRK1A, especially in the direction of the protein cavity [71]. In the case of GSK-3β, the pyridine ring of HAR interacts in the binding pocket with the hinge region of the kinase, while for DYRK1A, the methoxy group of HAR interacts with the hinge region. This difference may be the reason why HAR is a strong DYRK1A inhibitor. 

Suzuki coupling and Cadogan cyclization were used for the preparation of HAR and a series of fifteen derivatives, which were screened for dual GSK-3β/DYRK1A inhibitory activity. Among them, the carbonyl group as a hydrogen bond acceptor containing compound ZDWX-25 (systematically: 1-(cyclopropanecarboxamido)-9*H*-pyrido [3,4-b]indole-7-carboxylic acid methyl ester; Figure 3) showed potent inhibitory effects on GSK-3β and DYRK1A with IC_50_ values of 71 ± 9 nM and 103 ± 4 nM, respectively (Table 1). Molecular modeling and kinetic experiment confirmed that ZDWX-25 could interact with the ATP binding pocket of GSK-3β and DYRK1A [71]. It was also able to penetrate the BBB in vitro (*P_e_* = 16.5 × 10^−6^ cm/s; CNS+).

### 2.3. Protoberberine Alkaloids: Berberine and Palmatine

Berberine (BBR) and palmatine (PAL) are quaternary isoquinoline alkaloids of protoberberine-type, biosynthetically derived from tyrosine (Figure 4). Both alkaloids occur in Coptidis rhizoma and Corydalis rhizoma, traditional Chinese herbs used for memory enhancement [80].

BBR is commonly found and isolated from the roots, rhizomes, stems, and bark of *Coptis chinensis* Franch. (Ranunculaceae), *Berberis vulgaris* L. (Berberidaceae), *Hydrastis canadensis* L. (Ranunculaceae), and *Phellodendron amurense* Rupr. (Rutaceae). For decades, Chinese medicine has used plants and their extracts containing BBR to treat various diseases [96]. BBR was used in China as a folk medicine by Shennong at approximately 3000 BC, and the first recorded use of BBR is described in the ancient Chinese medical book *The Divine Farmer’s Herb-Root Classic* [97]. BBR shows a wide range of biological activities, including anti-viral, anti-bacterial, anti-inflammatory, anti-cancer, anti-hypoglycemic, and others [98]. For these reasons, it is one of the most studied natural products, which has so far been validated in about 77 clinical trials in different areas [99]. Moreover, BBR is one of the most reviewed natural products, and many of them have been published in the past ten years [100,101,102,103,104,105,106]; thus, we will summarize only the most important results in connection with the potential treatment of AD [107].

In recent years, BBR has been extensively investigated by various researchers for its activity against AD; numerous studies have indicated that BBR treatment significantly improves memory and cognitive dysfunction in different animal models of AD [104,107]. For example, intragastric administration of 50 mg/kg of BBR once daily for 14 days demonstrated a pronounced improvement in spatial memory deficits in a rat AD model [108], which was also proved in a study with streptozocin-diabetic rats in a dose of 100 mg/kg. Investigators attributed this effect to the restoration of synaptic plasticity and an anti-apoptotic property [109]. In another study, a daily dose of BBR (50 mg/kg) for three weeks in nonalcoholic steatohepatitis rats had neuroprotective effects on increased Aβ_42_ production, AChE activity, and inflammation [110]. In the next study, the intraperitoneal dosage of 20 mg/kg of BBR for two weeks also improved memory impairment in rats induced by scopolamine, as observed in passive avoidance and MWM tests [111]. Another study using rats with diabetes further demonstrated that BBR given orally (100 mg/kg) ameliorated learning and memory deficit due to the prevention of oxidative stress and ChE activity [112]. BBR in 3xTg-AD mice also improved their spatial learning ability and memory and was also proven to promote the autophagy clearance of Aβ by the class III phosphoinositide 3-kinase (PI3K)/beclin-1 pathway and inhibit its production by the inhibition of BACE-1 expression [113]. Using the same 3xTg-AD mice, BBR was also shown to promote the formation of microvessels by enhancing brain CD31, vascular endothelial growth factor, N-cadherin, and angiopoietin 1, which contributed to cerebral blood flow recovery [114]. BBR’s cognitive-enhancement effect was also measured in TgCRND8 mice, which were receiving either 25 or 100 mg/kg of BBR by oral gavage for four months. In this study, BBR was shown to reduce Aβ cerebral levels and glial activation; additionally, BBR suppressed APP levels via activation of the PI3K/protein kinase B (Akt)/GSK-3 signaling pathway in N2a mouse neuroblastoma cells [107]. Due to its neuroprotective capabilities, BBR was also shown to ameliorate doxorubicin-induced cognitive decline in rats, with studied underlying mechanisms comprised of attenuating expression of inflammatory proteins and genes, apoptotic factors Bax and Bcl2, up-regulating the expression of peroxisome proliferator-activated receptor-γ 1α and manganese superoxide dismutase, and overall, improving synaptic plasticity through cyclic adenosine monophosphate response element-binding protein and brain-derived neurotrophic factor [115]. In a rabbit model of AD simulated by an aluminum injection into the intraventricular fissure, the oral administration of BBR chloride (50 mg/kg) protected the rabbit hippocampus from degeneration and prevented the increased activity of BACE-1 by 40% [116]; however, the IC_50_ for BACE-1 was in another study determined to be higher than 100 μM [81]. BBR has also been shown to reduce the formation of Aβ and decrease the expression of BACE-1 by activating AMP-activated protein kinase in N2a mouse neuroblastoma cells [117]. The decrease in Aβ_40/42_ production by BBR was also confirmed by a study on HEK293 cells, which can be explained by the inhibition of the expression of BACE via activation of the extracellular signal-regulated kinase 1/2 pathway [118]. The study by Brunhofer et al. reported an IC_50_ for the inhibition of Aβ_1–40_ of 43.84 μM (Table 1), as well as an IC_50_ for the induction of Aβ_1–40_ disaggregation (104.90 μM) [82]. Additionally, BBR is capable of mitigating the hyperphosphorylation of tau protein by inhibiting the nuclear factor-κB pathway [119] and by inhibiting GSK-3β [120], further reducing the cognitive deficit in AD, which was also verified in a mouse model [120]. In HEK293 cells, BBR also reduced tau hyperphosphorylation induced by calyculin A [121]. Simultaneously, BBR is able to suppress neuroinflammation by decreasing the production of TNF-α and interleukin-1β (IL-1β) [122], as well as acting as an antioxidant by influencing the PI3K/Akt signaling pathway [123]. It is noteworthy to mention that BBR has significant cholinesterase inhibitory activity, as shown in Table 1. For *h*AChE, its IC_50_ ranges from 0.52 μM to 0.7 μM with *K*_i_ = 0.54 μM [80,83]. Compared to BuChE inhibition, BBR seems to be selectively active towards AChE, since its IC_50_ values are several times higher, being up to 30.7 μM for *h*BuChE inhibition [83]. BBR can also inhibit MAO-A enzyme with an IC_50_ of 126 μM [84]. BBR, according to in vitro PAMPA studies, seems to be unable to cross the BBB [83,85]; however, Wang et al. detected that BBR is able to accumulate in the hippocampus of rats after the intravenous injection of Coptis rhizoma extract [124]. BBR has a safe, non-toxic profile and can be administered orally [105]. Lastly, BBR is capable of decreasing the role of AD’s risk factors, such as atherosclerosis and diabetes [103].

PAL can be found in the roots of *Coptis chinensis*, and *Corydalis* DC. species (Papaveraceae), and many other herbs used in traditional Chinese medicine [125,126,127]. Studies have shown that PAL is a potent inhibitor of AChE, displaying IC_50_ values ranging from 0.46 μM [80] to 1.69 μM for *h*AChE [82]; however, the results for BuChE inhibition vary depending on the source of the enzyme (IC_50_ for *h*BuChE > 100 μM [83], and 6.84 μM for *eq*BuChE; Table 1). Moreover, PAL demonstrated an interesting inhibitory activity against MAO-A with an IC_50_ = 47 μM [87]. PAL was tested in vivo on Swiss albino mice, showing memory-enhancing activity from a concentration of 0.5 mg/kg in the MWM test. In a concentration of 1 mg/kg, PAL also substantially reversed amnesia induced by scopolamine and diazepam [128]. Furthermore, 7-day administration of PAL in a concentration of 10 mg/kg to 12-month-old 5xFAD mice significantly improved learning and memory tested by the MWM test. PAL is able to penetrate the BBB, as multiple reaction monitoring analyses revealed. According to data by Kiris et al., PAL is capable of causing changes in the cerebellum and hippocampus, but not in the brain cortex [129]. The neuroprotective effect of PAL was studied in vivo on the *C. elegans* AD model containing human Aβ_1–42_, showing that treatment significantly delayed the paralytic process, reduced the amount of oxidative stress, and alleviated the deposition of Aβ [130]. The capability of anti-neuroinflammatory potential was also previously studied, demonstrating PAL’s ability to inhibit the production of several inflammatory mediators such as NO, reactive oxygen species, and matrix metallopeptidase 9 in microglia BV-2 cells [131].

The study of Mak et al. elucidated the effect of simultaneous administration of BBR and PAL on the inhibition of *h*AChE in vitro, revealing the synergic action of these two protoberberine alkaloids [80]. The high activity of those alkaloids could be attributed to the positively charged nitrogen, which can bind to the gorge of the active site of AChE [132]. PAL and BBR are chemically similar, except for the pattern of substitutions on the dihydroisoquinoline structure. Dioxymethylene substitution is preferred to vicinal dimethoxy substituents, as shown by the data of Brunhofer et al. [82]. Overall, the studies suggest that BBR and PAL have immense therapeutic potential in treating AD, although further research is needed to fully assess and understand their effects.

### 2.4. Benzophenathridine Alkaloids: Avicine, Nitidine, and Chelerythrine

Preliminary screening studies led to the selection of *Zanthoxylum rigidum* Humb. et Bonpl. ex Willd. (Rutaceae) for detailed phytochemical study [86]. Multi-step chromatography of root extract yielded various alkaloids, including two benzophenathridine structures, avicine and nitidine, which were tested for *Ee*AChE, *h*AChE, and *eq*BuChE inhibitory activity, as well as for MAO-A and B inhibition, and Aβ aggregation. Avicine contains in its structure a dioxomethylene bridge connecting positions 8 and 9, while in nitidine, these positions are substituted by two methoxy groups (Figure 5). Both compounds showed dual cholinesterase inhibition activity, being more active against AChE than BuChE, with IC_50_ values in the (sub)micromolar concentration range (Table 1). Both alkaloids showed increased inhibition towards *Ee*AChE compared with *h*AChE. Moreover, avicine also demonstrated significant *eq*BuChE activity (IC_50_ = 0.88 ± 0.08 µM), with an *eq*BuChE/*h*AChE selectivity index of 1.67. Kinetic studies indicated that avicine and nitidine are reversible-mixed inhibitors of both cholinesterases. Mixed-type inhibitors are able to bind at the catalytic and peripheral anionic sites (PAS) of the enzyme. PAS inhibitors can regulate ChE-induced Aβ aggregation, which supported results obtained within the study, since avicine and nitidine were able to inhibit Aβ_1–42_ aggregation with IC_50_ values of 5.56 ±0.94 µM and 1.89 ± 0.40 µM, respectively [86]. In the MAO inhibition assay, both alkaloids demonstrated inhibition potency against isoform A of human recombinant MAO in micromolar concentrations (Table 1) but were inactive against MAO-B (IC_50_ > 100 µM).

A further benzophenanthridine alkaloid chelerythrine, commonly isolated from *Chelidonium majus* L. (Papaveraceae) [133], contains the same structural motif as nitidine. The two alkaloids differ only in the position of one methoxy group on aromatic ring A (Figure 5). Chelerythrine also demonstrated dual cholinesterase activities. Interestingly, it showed slightly higher activity towards the *h*AChE (1.54 ± 0.07 µM) than towards the *Ee*AChE (3.78 ± 0.15 µM). A reversed situation was obtained for BuChE, as chelerythrine was slightly more active towards the horse *eq*BuChE (6.33 ± 0.95 µM) when compared to the human enzyme (10.34 ± 0.24 µM; Table 1). Kinetic studies were performed on *Ee*AChE and *h*AChE; the kinetic curves revealed that chelerythrine is a mixed-type inhibitor of *Ee*AChE, with a slightly higher competitive behavior (*K_ic_* = 0.48 ± 0.07 µM and *K_ic_* = 0.92 ± 0.11 µM). The same mechanism of inhibition was found using *h*AChE with a *K_ic_* = 0.32 ± 0.08 µM and *K_ic_* = 1.12 ± 0.07 µM. Based on these results, a docking study of chelerythrine was performed using crystal structure PDB ID: 1FSS (from *Torpedo californica*) as a model receptor, as this structure is fundamentally similar to that of human AChE [82]. Chelerythrine covers the gorge of the active site showing a hydrogen bond interaction with Tyr130, as well as π-stacking interactions with Tyr121 and Tyr334, which are PAS residues. The identification of chelerythrine as a mixed-type inhibitor, as well as the results of the docking study, indicated that the compound may also inhibit AChE-induced Aβ aggregation. Compounds are regarded as good inhibitors of AChE-induced Aβ fibril formation if they show biological activity in the range of 82–98% at a concentration of 100 µM [134]. Chelerythrine inhibited AChE-induced Aβ aggregation at 5, 10, and 100 µM with 48.5%, 65.0%, and 88.4%, which indicate that chelerythrine can be recognized as a potent inhibitor of AChE-induced Aβ_1–40_ aggregation [82]. The influence of chelerythrine on Aβ_1–40_ aggregation was studied using 1,1,1,3,3,3-hexafluoro-2-propanol as an aggregation enhancer [135]. Chelerythrine demonstrated 67% inhibition of Aβ_1–40_ aggregation at a concentration of 10 µM; thus, the value of IC_50_ was subsequently determined (4.2 ± 0.43 µM), which indicated that chelerythrine is a highly active inhibitor of Aβ_1–40_ aggregation. Moreover, chelerythrine was also tested for its ability to disaggregate already preformed Aβ_1–40_ aggregates. This ability could be more relevant from a clinical perspective, when disaggregation of already existing Aβ fibrils in the AD brain could be indicated, especially at the beginning of the treatment to reduce the neurotoxic effects of Aβ fibrils and thus prevent neurodegeneration. Chelerythrine showed a high activity in disaggregating preformed Aβ_1–40_ aggregates, with an IC_50_ of 13.03 ± 2.89 µM after 45 min incubation [82].

Chelerythrine also selectively inhibited an isoform of recombinant human MAO-A with an IC_50_ value of 0.55 ± 0.042 µM [88] and was recognized as a reversible competitive MAO-A inhibitor (*K_i_* = 0.22 ± 0.033 µM). Docking simulation showed that chelerythrine binds to MAO-A due to two hydrogen bond interactions with Cys323 and Tyr444 [88].

## 3. Marine Alkaloids and Nitrogen Containing Compounds as Multi-Target Compounds for the Treatment of AD

Almost all of the current natural product-derived therapeutics have terrestrial origins. However, there are four approved marine or marine-derived drugs (cytarabine, vidarabine, trabectedin, and eribulin mesylate), which are used as either anti-viral or anticancer agents (Figure 6) [136,137]. Thus, marine natural products are proven to be an important source of various structural scaffolds for developing novel drugs with a wide range of biological properties such as anti-bacterial, anti-viral, anti-tumor, anti-inflammatory, and neurological activities [138,139,140].

### 3.1. Imidazole Alkaloids: Pseudozoanthoxanthin and Stevensine

An ethanolic extract of a zoanthid crust coral, *Parazoanthus axinellae*, exhibited anticholinesterase activity. Subsequent RP-HPLC separation led to the isolation of pseudozoanthoxanthin (PSX), containing 2-amino imidazole groups in its structure (Figure 7), which has been recognized as a competitive inhibitor of AChE with a *K_i_* = 4 µM [141]. Whereas the 2-amino imidazole group has been identified in known BACE-1 inhibitors [142], this alkaloid has been repetitively isolated from an unidentified Caribbean coral collected in Mexico for more detailed studies [143]. For these advanced studies, the bromo-pyrrole alkaloid stevensine (STV, Figure 7) has also been isolated from the sponge *Axinella verrucose* collected in the Gulf of Naples. Both alkaloids were predicted by a computational approach to possess interesting multitarget profiles on AD target proteins, which have been confirmed by in vitro experiments. The inhibitory activity of PSX and STV was evaluated using *h*AChE and *eq*BuChE. PSX showed moderate inhibition activity against both tested alkaloids, while STV was a moderate inhibitor of *h*AChE and a weak inhibitor of *eq*BuChE (Table 2). Docking studies carried out on human BACE-1 protein with the flap loop conformation (PDB ID: 2QZL) showed that the amino-imidazole group is able to stably interact with both the catalytic aspartases and with the close Thr residue, whereas the rest of the molecules interacts with the flap loop that actively contributes to the overall binding of these molecules. Murine BACE-1, which shares 95% sequence identity with *h*BuChE, has been used for the validation of the inhibitory properties of PSX and STV. Both compounds strongly inhibited murine BACE-1 and showed similar IC_50_ values (Table 2) and were in agreement with predicted binding energies (−8.04 ± 0.02 kcal/mol for PSX and −8.89 ± 0.02 kcal/mol for STV). Preventing the early phases of Aβ amyloidogenesis is regarded as a promising therapeutic strategy, since it represents a crucial step toward the formation of neurotoxic oligomers [144,145]. The Aβ_1–40_ and Aβ_1–42_ anti-aggregation potential of PSX and RES was studied using ThT fluorescence assay. Both alkaloids at the concentration of 50 µM caused significant inhibition of fibrillation of both peptides (Table 2). Moreover, the intensity of ThT fluorescence in the presence of Aβ_1–40_ and Aβ_1–42_ was reduced when co-incubated with PSX and STV, which indicated that both compounds are able to induce partial disaggregation of Aβ_1–40_ and Aβ_1–42_ complexes [143]. Further in vitro and pilot in vivo experiments have been reported only with PSX since available amounts of STV prevented further testing. In order to clarify the underlying mechanism of PSX on Aβ_1–42_ aggregation, high-resolution atomic force microscopy was used, which allowed deeper insight into the interaction process. Aβ_1–42_ protein is initially monomeric, but after reconstitution in buffer solution it starts the aggregation process, which is well studied and described [146]. These experiments show that soon after solubilization Aβ_1–42_ has a very strong tendency to aggregate and form oligomeric aggregates with a size of about 3 nm. PSX can interfere with this process by somewhat capping Aβ_1–42_ molecules and preventing further aggregation. Preliminary in vivo experiments were carried out to evaluate the pharmacological effects of PSX on attention, learning, working and spatial memory with respect to cortical and hippocampal electroencephalogram (EEG) theta rhythm during a cognitive performance in an experimental model of AD [147,148]. PSX was able to revert EEG and cognitive alterations induced by the nucleus basalis of Meynert excitotoxicity, thus recovering the cortical–hippocampal functional connectivity in mice.

Recent trends in therapeutic research of AD have considered the search for disease-modifying drugs that interfere with the pathology of Aβ/or tau phosphorylation. Tau phosphorylation is caused by the effects of different protein kinases and phosphatases. Thus, the blockade of this hyperphosphorylation step by selective inhibitors of tau kinases may be a prime site at which to interrupt the pathogenic cascade. Hymenialdisine (HD, Figure 6), which can be found in species of marine sponges belonging to the *Aelasidae*, *Axinellidae*, and *Halichondriade* families, has been identified as an inhibitor of various kinases within marine alkaloids carrying an imidazole-core in the structure [151]. HD has been tested on a variety of highly purified kinases [151]. Most kinases tested were either poorly or not inhibited (IC_50_ > 1 µM); however, four, CDK1/cyclin B, CDK5/p35, GSK-3β, and casein kinase 1 (CK1), were strongly sensitive to HD (IC_50_ values of 10 and 35 nM, respectively; Table 3). The HD-sensitive kinases were also assayed in vitro with physiologically relevant substrates: a fragment of presenilin-2 for CK1, Pak1 for CDK5/p35, and either the insulin-receptor substrate IRS-1, or tau for GSK-3β. The sensitivity of the kinases towards HD was comparable to those of the same kinases assayed with more artificial substrates.

### 3.2. Indole Alkaloids: Meridianins

Meridianins are a family of indole alkaloids isolated from marine benthic organisms from Antarctica [153,154]. These ascidian alkaloids consist of an indole framework linked to an aminopyrimidine ring. Like HD, some of these compounds have been identified as potent inhibitors of various kinases (Table 3) [153]. Docking calculations and molecular dynamic simulations showed the ability of meridianins to act as either ATP-competitive or non-ATP-competitive inhibitors of GSK-3β [155]. The same study demonstrated the capacity of meridianins to inhibit GSK-3β in vitro without altering neuronal survival. A further study examined whether a mixture of meridianins was capable of inhibiting neural GSK-3β in vivo, and if such inhibition induces improvement in the 5xFAD mouse model of AD [156]. It was found that the mixture of meridians induces structural synaptic plasticity in primary hippocampal neurons and that their intracerebral administration in living mice inhibits GSK-3β in the hippocampal region. Direct administration of meridianins in the third ventricle of 5xFAD mice induced robust improvements in recognition memory and cognitive flexibility, as well as a rescue of the synaptic loss and amelioration of neuroinflammatory processes. Unfortunately, meridianins do not cross the BBB; thus, future studies should be carried out with meridianin-based synthetic compounds [156].

### 3.3. Further Nitrogen Containing Marine Compounds as Multi-Target Compounds for the Treatment of AD: Pulmonarin B

Pulmonarin B (PLMB), a dibrominated phenylacetic acid derivative containing a quaternary ammonium group, has been isolated from the ascidian *Synoicum pulmonaria* by Sevenson et al. and screened for its *Ee*AChE inhibition potency [157]. In this pilot study, PLMB has been identified as a reversible, non-competitive AChE inhibitor with an inhibition constant (*K_i_*) of 20 µM. In a follow-up study, PLMB showed balanced inhibitory activity against both cholinesterases with IC_50_ values of 37.02 ± 2.11 µM for AChE and 30.70 ± 1.44 µM for BuChE [150]. PLMB was subsequently studied for its potential to inhibit self-induced and AChE-induced Aβ_1–42_ aggregation using the ThT fluorescence method. Tacrine and donepezil were used as positive control. PLMB is a weaker inhibitor of self-induced Aβ_1–42_ aggregation (29.78 ± 1.45% at 10 µM) compared to donepezil. On the other hand, PLMB showed comparable potency to inhibit AChE-induced Aβ_1–42_ aggregation (27.60 ± 1.96% at 10 µM) as donepezil (22.42 ± 2.56% at 10 µM) [150]. Moreover, PLMB showed no cytotoxicity against the HepG2 cell line (IC_50_ > 80 µM).

### 3.4. Quinazoline-Benzodiazepine Alkaloid: Circumdatin D

Circumdatin D (CIRD, Figure 7) is a quinazoline-benzodiazepine alkaloid originally isolated from a terrestrial strain of the fungus *Aspergillus ochraceus*, together with other circumdatins [158], and subsequently found in other strains of the species associated with marine brown algae [159] or gorgonian coral [160]. This compound has been shown to possess several biological activities that may ameliorate the pathophysiology of AD. In an AChE inhibition assay, CIRD was the most active of all isolated circumdatins with an IC_50_ value of 2.4 ± 0.5 µM, and thus it was selected for further experiments. In vitro, CIRD has demonstrated neuroprotective effects by inhibiting the toll-like receptor 4-mediated NF-κB, mitogen-activated protein kinases, and Janus kinase/signal transducer and activator of transcription protein signaling pathways responsible for neuroinflammation. Furthermore, this alkaloid has been shown to substantially inhibit the lipopolysaccharide (LPS)-induced production of NO and cytokines such as TNF-α, IL-1β, and cyclooxygenase-2 expression in microglial BV-2 cells. CIRD-treated neuronal cells have also been observed to have significantly reduced LPS-induced AChE activity. The AD model of *C. elegans* strain CL4176 was used for in vivo assay of CIRD. This model expresses Aβ in muscle cells, which leads to progressive neurodegeneration and paralysis [161]. Moreover, further pathological features include AChE and inflammation gene overexpression. The study revealed that CIRD markedly reduced the paralysis of the nematodes upon temperature up-shift compared to untreated animals. CIRD also significantly inhibited the AChE activities, and reduced the expression of inflammatory genes in *C. elegans* [160].

## 4. Conclusions

Multi-factorial diseases like AD require complex treatment strategies that involve simultaneous modulation of a network of interacting targets. During the last few years, the multi-target compounds have been explored as an effective therapeutic approach for the treatment of AD. Besides direct inhibition of enzymes involved in AD pathology, intensively studied targets include amyloid plaque deposition, neuroinflammation signaling pathways, anti-apoptotic and anti-oxidative stress activities, and neuroprotection.

In the current review article, which is focused on alkaloids and other nitrogen compounds, various pathogenic pathways associated with AD have been briefly described, followed by an overview of selected natural alkaloids that can be recognized as multi-target compounds for the development of new anti-AD drugs. Of these, harmine is the most promising alkaloid, displaying a wide spectrum of compelling anti-AD activities. Isoquinoline alkaloids such as berberine, avicine, and chelerythrine also appear to be promising multi-target compounds, exhibiting strong inhibitory activity on key pathological enzymes of AD. Furthermore, marine flora have emerged as a viable source of multi-target compounds as well; for example, hymenialdisine has a broad range of protein kinase-inhibiting activities in a nanomolar range. Such compounds are currently of interest for the preparation of semi-synthetic derivatives.

To conclude, it is essential to undertake more studies on the discussed alkaloids at a cellular and molecular level, such as influence on the various signal pathways connected to the neurodegeneration, docking studies and molecular dynamics studies in the active sites of the target enzymes, and cytotoxicity evaluations to select clinically important candidates for treating neurodegenerative diseases.

## Figures and Tables

**Figure 1 ijms-24-04399-f001:**
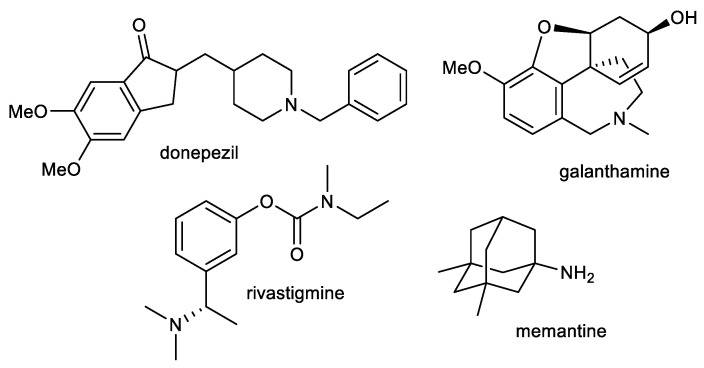
Structures of available drugs for treatment of AD.

**Figure 2 ijms-24-04399-f002:**
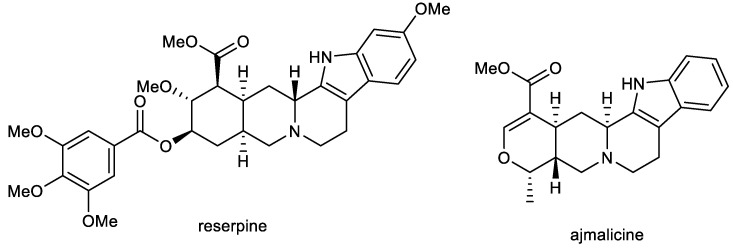
Structures of reserpine and ajmalicine.

**Figure 3 ijms-24-04399-f003:**
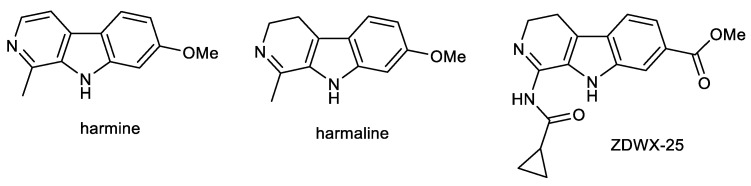
Structures of β-carboline alkaloids harmine, harmaline and derivative ZDWX-25.

**Figure 4 ijms-24-04399-f004:**
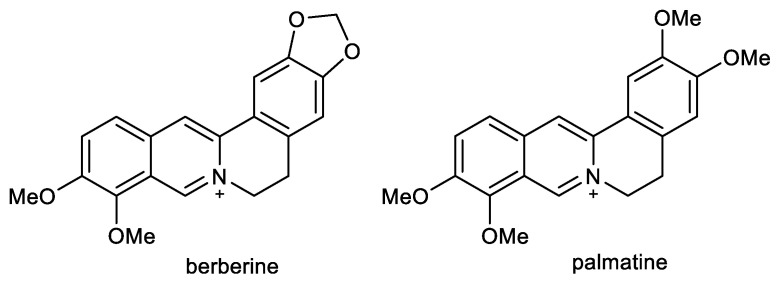
Structures of berberine and palmatine.

**Figure 5 ijms-24-04399-f005:**
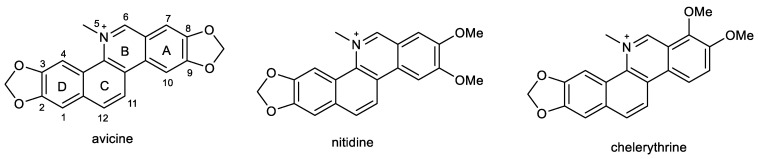
Structures of discussed benzophenanthridine alkaloids.

**Figure 6 ijms-24-04399-f006:**
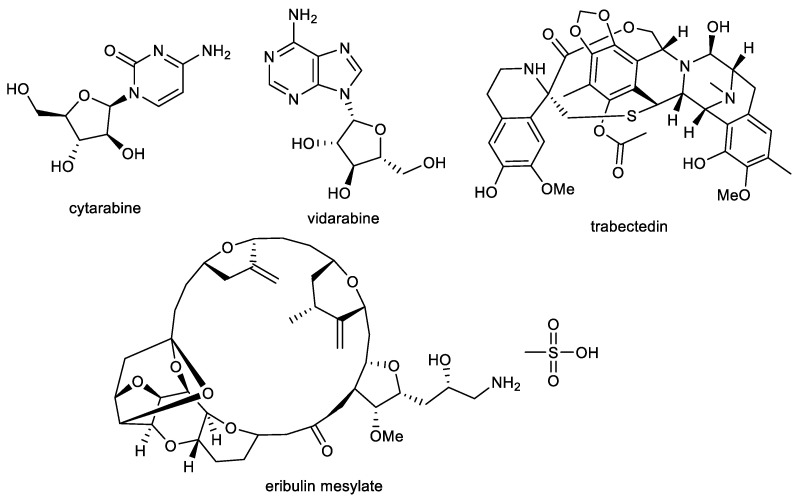
Structures of approved marine or marine-derived drugs.

**Figure 7 ijms-24-04399-f007:**
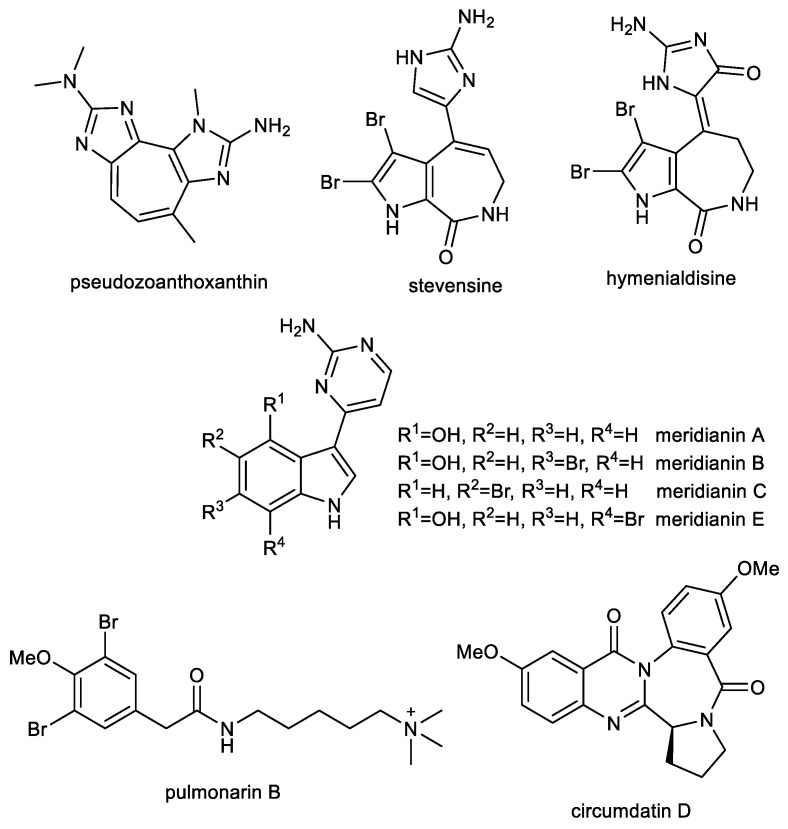
Structures of discussed marine alkaloids and nitrogen compounds.

**Table 1 ijms-24-04399-t001:** Biological activity of selected plant alkaloids in connection with AD.

Compound	AChE IC_50_ (µM)	BuChE IC_50_ (µM)	BACE-1 (% Inhibition at Given Conc.)	Aβ_1–42_ IC_50_ (µM)	MAO-A IC_50_ (µM)	CDK5 IC_50_ (µM)	GSK-3β IC_50_ (µM)	DYRK1A IC_50_ (µM)	PAMPA BBB Permeation *P_e_* (×10^−6^ cm/s)	Ref.
Harmine	1.21 ± 0.04 ^b^; 9.05 ± 1.08 ^b^	2.79 ± 0.27 ^d^; 75.07 ± 1.29 ^d^	0 ± 13 at 10 µM	2.5 ± 0.7	0.38 ± 0.21	20	31.1 ± 1.0	0.080 ± 0.007	44.6 (CNS+); 7.17 ± 1.29 (CNS+)	[70,71,72,73,74,75,76]
Harmaline	1.95 ± 0.08 ^b^; 10.58 ± 2.01 ^b^	5.38 ± 0.64 ^d^; 101.39 ± 1.39 ^d^	n.d.	n.d.	0.10 ± 0.08	Inhibition 7% at 50 µM	n.d.	9	n.d.	[70,72,76,77,78]
ZDWX-25	n.d.	n.d.	n.d.	n.d.	n.d.	n.d.	0.071 ± 0.009	0.103 ± 0.004	16.5 (CNS+)	[71]
Reserpine	1.7 ± 2.08 ^b^	2.8 ± 1.84 ^d^	47 at 50 µM	57% at 44 µM	Increase in activity	n.d.	n.d.	n.d.	2.75 ± 0.25 (CNS+/−)	[68,75,79]
Ajmalicine	3.5 ± 1.41 ^b^	5.44 ± 1.75 ^d^	69 at 50 µM	69% at 44 µM	n.d.	n.d.	n.d.	n.d.	n.d.	[68]
Berberine	0.520 ± 0.042 ^a^; 0.61 ± 0.04 ^a^; 0.7 ± 0.1 ^a^; 0.44 ± 0.04 ^b^; 2.74 ± 0.22 ^b^	30.7 ± 3.5 ^c^; 6.40 ± 0.29 ^d^; 3.44 ± 0.26 ^d^	IC_50_ > 100 µM	43.84 ± 6.09	126	n.d.	n.d.	n.d.	0.1 ± 0.1 (CNS−); 0.02 (CNS−)	[80,81,82,83,84,85,86]
Palmatine	0.46 ± 0.013 ^a^; 1.69 ± 0.11 ^a^; 4.07 ± 0.09 ^b^; 0.51 ± 0.00 ^b^	>100 ^c^; 6.84 ± 0.07 ^d^	IC_50_ > 100 µM	92.15 ± 3.42	63.86 ± 1.35	n.d.	n.d.	n.d.	0 (CNS−)	[80,81,82,83,85,87]
Avicine	0.52 ± 0.05 ^a^; 0.15 ± 0.01 ^b^	0.88 ± 0.08 ^d^	n.d.	5.56 ± 0.94	0.41 ± 0.02	n.d.	n.d.	n.d.	n.d.	[86]
Nitidine	1.25 ± 0.09 ^a^; 0.65 ± 0.09 ^b^	5.73 ± 0.60 ^d^	n.d.	1.89 ± 0.40	1.89 ± 0.17	n.d.	n.d.	n.d.	n.d.	[86]
Chelerythrine	1.54 ± 0.07 ^a^; 1.03 ± 0.11 ^b^	10.34 ± 0.24 ^c^; 3.55 ± 0.18 ^d^	n.d.	4.20 ± 0.43	0.55 ± 0.042	n.d.	0% at 10 µM	n.d.	3.25 (CNS+/−)	[82,85,86,88,89]

^a^*h*AChE, ^b^*Ee*AChE, ^c^*h*BuChE, ^d^*eq*BuChE, n.d. = not determined.

**Table 2 ijms-24-04399-t002:** Biological activity of selected marine alkaloids in connection with AD.

Compound	AChE IC_50_ (µM)	BuChE IC_50_ (µM)	BACE-1 IC_50_ (µM)	Aβ_1–42_ (% Inhibition at Given Conc.)	DYRK1A IC_50_ (µM)	Ref.
Pseudozoanthoxanthin	12.2 ± 1.4 ^a^	14.6 ± 5.4 ^c^	0.9 ± 0.1	50% at 50 µM	n.d.	[143]
Stevensine	7.8 ± 1.5 ^a^	141.6 ± 34.0 ^c^	1.4 ± 0.4	n.d.	n.d.	[143]
Hymenialdisine	n.d.	n.d.	n.d.	n.d.	0.0033	[149]
Pulmonarin B	37.02 ± 2.11 ^b^	30.70 ± 1.44 ^c^	n.d.	29.78 ± 1.45% at 10 µM	n.d.	[150]

^a^*h*AChE, ^b^*Ee*AChE, ^c^*eq*BuChE, n.d. = not determined.

**Table 3 ijms-24-04399-t003:** Effects of hymenialdisine and meridianins on the activity of selected protein kinases.

Compound	CDK1 IC_50_ (µM)	CDK5 IC_50_ (µM)	GSK-3β IC_50_ (µM)	CK1 IC_50_ (µM)	Ref.
Hymenialdisine	0.022	0.028	0.010	0.035	[152]
Meridianin A	2.50	3.00	1.30	n.d.	[153]
Meridianin B	1.50	1.00	0.50	1.00	[153]
Meridianin C	3.00	6.00	2.00	30.00	[153]
Meridianin E	0.18	0.15	2.50	100.00	[153]

n.d. = not determined

## Data Availability

Data sharing not applicable.

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
