# Peer review of "Natural Alkaloids as Multi-Target Compounds towards Factors Implicated in Alzheimer’s Disease"

_ijms, 2023, doi:10.3390/ijms24054399_

Round 1
Reviewer 1 Report
The manuscript reviews literature data on alkaloids functioning as multi-target-directed ligands towards various factors involved in AD (AChE, BuChE, BACE-1, MAO-A, Abeta1-42, CK1, CDK1, CDK5, GSK-3beta, DYRK1A).
The review offers a comprehensive and in-depth overview on this topic. The molecular mechanisms underlying the multi-target type activity of alkaloids, even though very complex, are clearly presented. The writing style is very clear, easy to read and understand despite the complexity of the information.
In my opinion, the manuscript has high scientific quality and represents a valuable current status of knowledge in the field. Therefore, I recommend acceptance in the present form.
Author Response
Dear Editor,
we have carefully revised the manuscript entitled “Natural Alkaloids as Multi-Target-Directed Ligands Towards Factors Implicated in Alzheimer's disease“ Manuscript ID: ijms-2210183, according to the reviewers' suggestions. All changes are highlighted in red and are visible in the MS Word’s “Track Changes” function.
Comments from the reviewers:
Reviewer #1:
The manuscript reviews literature data on alkaloids functioning as multi-target-directed ligands towards various factors involved in AD (AChE, BuChE, BACE-1, MAO-A, Abeta1-42, CK1, CDK1, CDK5, GSK-3beta, DYRK1A).
The review offers a comprehensive and in-depth overview on this topic. The molecular mechanisms underlying the multi-target type activity of alkaloids, even though very complex, are clearly presented. The writing style is very clear, easy to read and understand despite the complexity of the information.
In my opinion, the manuscript has high scientific quality and represents a valuable current status of knowledge in the field. Therefore, I recommend acceptance in the present form.
Thank you very much for this positive evaluation of our work.
Reviewer #2:
Good review with useful information on the subject matter. I would suggest a couple things:
- That the authors Place figures 2-5 &b 7 directly under their sectional subheadings. This is simply for readability.
We thank the reviewer for this suggestion that improves the readability and have adjusted the placement of the figures accordingly.
- That the authors also include the recently approved Lecanemab (Leqembi) on page 4.
We thank the reviewer for notification about this very recently approved drug, which is now also included in our manuscript.
Reviewer #3:
The present review aims to comprehensively summarize alkaloids of various origins as multi-target-directed ligands for acetylcholinesterase (AChE).
In the first section of the review, authors discuss about the molecular pathology of AD. In the following section, authors comprehensively summarize the research that has been published on selected natural alkaloids as potential multi-target-directed ligands towards factors implicated in AD. The authors also presents studies involving different types of alkaloids and i two dimensional molecular structure. In my opinion, the major highlight of the review is the data presented in Table 1. Biological activity of selected plant alkaloids in connection with AD is an important resource compiled from literature which is an excellent addition to the literature. In section 3, the review deals with marine alkaloids and nitrogen containing compounds as multi-target ligands for the treatment of AD. In my opinion, this review is well written and an important addition to the literature.
Thank you very much for this positive evaluation of our work.
Before accepting this paper, I have couple of minor concerns to be addressed:
The authors concluded that, “it is essential to undertake more studies on the discussed alkaloids at a cellular and molecular level to select clinically important candidates for treating neuro-degenerative diseases.” I request the authors to expand the statement in detail about what kind of studies can be undertaken by the researchers worldwide and how the authors group addresses this issue?
We thank the reviewer for this remark and we have incorporated this information in the manuscript.
The limitation of this review should be mentioned in abstract/conclusion section.
We have incorporated the limitation statement in the conclusion section.
Reviewer #4:
The manuscript entitled “Natural Alkaloids as Multi-Target-Directed Ligands Towards Factors Implicated in Alzheimer’s disease” described pleiotropic activities of alkaloids in the context of neurodegenerative processes, mainly related to cholinergic dysfunction, oxidative stress, Aβ and τ pathways. The main topic still remains hot in medicinal chemistry, considering that natural compounds such as alkaloids have represented the main source of inspiration in drug development since ever. The review is well written and comprehensive, sometimes even too detailed, but presents significant flaws, two major and few minor, that have to be stressed.
- Novelty. Alkaloids, starting from cited galantamine and reserpine, have always represented a resource for medicine first and now mainly a starting point for drug design. Therefore, in the literature there are plenty of research articles regarding alkaloids’ SAR as well as reviews summarizing them. Among these, a lot of reviews have been published in last years on alkaloids’ neuroprotective properties and some of them are cited in the manuscript but many are not:
- 2020. Current Medicinal Chemistry, 2020, 27, 5887-5917
- 2021. Bioorganic & Medicinal Chemistry 43 (2021) 116270
- 2021. Biomedicine & Pharmacotherapy 139 (2021) 111609
- 2021. Molecules 2021, 26, 728.
Herein there are reported only four examples, but there are even more, beside the comprehensive book chapter of 2018 (cited, https://doi.org/10.1016/B978-0-444-64183-0.00008-7). Not all are the same, some reviews are more general (i.e., natural products) and other more specific (i.e., spirocyclic, chemical-based) but in this reviewer’s opinion there is no need of another alkaloid’s review, even if this manuscript in some paragraph is more specific and detailed.
We thank the reviewer for pointing out some of the missing literature which should have been cited. We have incorporated these papers in our review article.
- Nomenclature. The development of Multi-Target-Directed Ligands refers to the design strategy (or approach) where two (or more) pharmacophores are combined (linked, merged, fused, ….) with resulting polyhedral/synergistic biological outcome due to simultaneous interactions of pharmacophores with the “respective” targets. In parallel, there are privileged structures, such as usually happen for natural compounds, able per se to interact with multiple targets, therefore exhibiting a multitarget profile. In this case, there are no distinct pharmacophores but a same structure that can exert a pleiotropic activity. As a result, it is not correct to recall natural compounds as MTDLs, where multitarget compounds sounds more appropriate. Please modify accordingly along the manuscript.
We thank the reviewer for the opinion about the nomenclature of MTDLs. The title and the content of the manuscript were discussed with the invited editor as soon as we received the invitation to submit a manuscript for this Special Issue of IJMS.
Moreover, we are of the opinion that the discussed substances can be considered as multi-target compounds due to the fact that, for individual activities, the details of the molecule responsible for the given biological activity have not yet been identified for the discussed substances, as is known, for example, for galanthamine. That is that one part of the molecule may be responsible for e.g. inhibition of AChE/BuChE and another part for another biological activity. To reveal these structural details, a wide range of derivatives would need to be tested. Considering that some derivatives are found in plants only in trace amounts and their isolation in sufficient quantities would be difficult, it is probably not realistic to carry out these SAR studies.
- The introduction section is too detailed and, in some part, heavy to read, because there are some details that are not essential for the reader which makes it difficult to read (e.g., section regarding different AChE enzymes). Besides the importance of introducing first all the involved potential targets, there is no need of 5 pages extensively describing etiopathology of AD. It is enough to present each interested target and their networks without describing in detail all the pathological cascades. The same happens in the rest of the manuscript where too details are likely to weaken reader’s attention and lose key points. It is better to highlight most important activities and only cite the other biological properties.
We thank the reviewer for the insight; however, we feel that the extensive description of the pathophysiology of AD is necessary for the readers to fully understand the meaning and limitations of the summarized activities.
- Please correct the reference section because they are numbered twice.
Corrected.
Based on these premises, especially the first point, I suggest to not accept this manuscript for publication on International Journal of Molecular Sciences.
We respect the reviewer’s point of view about our manuscript and thank him for his time evaluating our work.
Reviewer #5:
A very good paper. A little verbose in the présentation of the pathology of AD, could have been a little more concise, nevertheless a very welcome refresher for readers less clear on the mechanism and pharmacology of AD. The pharmacognostic part however is a pleasure to read. Well documented and clear enough.
Thank you very much for this positive evaluation of our work.
It could have benefited by a summarizing table and perhaps a visual aid like maybe a Venn diagram of overlapping mechanisms for various plant components, but it is just an opinion.
Thank you for providing this idea which could improve the quality of our paper; however, we feel that such a diagram or other visual aid is not necessary in our case, since we summarized relatively narrow spectrum of compounds.
It can be published as it is. Very serious work.
Thank you for the positive feedback once again.
In light of these changes, we are positive that our revised manuscript meets the criteria to be published in the International Journal of Molecular Sciences and would be of interest to all readers from the scientific community.
On behalf of all authors
Yours Sincerely,
Lucie Cahlíková, prof.

Reviewer 2 Report
Good review with useful information on the subject matter. I would suggest a couple things:
1. That the authors Place figures 2-5 &b 7 directly under their sectional subheadings. This is simply for readability.
2. That the authors also include the recently approved Lecanemab (Leqembi) on page 4.
Author Response

(The authors gave the same response as above.)

Reviewer 3 Report
The present review aims to comprehensively summarize alkaloids of various origins as multi-target-directed ligands for acetylcholinesterase (AChE).
In the first section of the review, authors discuss about the molecular pathology of AD. In the following section, authors comprehensively summarize the research that has been published on selected natural alkaloids as potential multi-target-directed ligands towards factors implicated in AD. The authors also presents studies involving different types of alkaloids and its two dimensional molecular structure. In my opinion, the major highlight of the review is the data presented in Table 1. Biological activity of selected plant alkaloids in connection with AD is an important resource compiled from literature which is an excellent addition to the literature. In section 3, the review deals with marine alkaloids and nitrogen containing compounds as multi-target ligands for the treatment of AD. In my opinion, this review is well written and an important addition to the literature.
Before accepting this paper, I have couple of minor concerns to be addressed:
The authors concluded that, “it is essential to undertake more studies on the discussed alkaloids at a cellular and molecular level to select clinically important candidates for treating neuro-degenerative diseases.” I request the authors to expand the statement in detail about what kind of studies can be undertaken by the researchers worldwide and how the authors group addresses this issue?
The limitation of this review should be mentioned in abstract/conclusion section.
Author Response

(The authors gave the same response as above.)

Reviewer 4 Report
The manuscript entitled “Natural Alkaloids as Multi-Target-Directed Ligands Towards Factors Implicated in Alzheimer’s disease” described pleiotropic activities of alkaloids in the context of neurodegenerative processes, mainly related to cholinergic dysfunction, oxidative stress, Aβ and τ pathways. The main topic still remains hot in medicinal chemistry, considering that natural compounds such as alkaloids have represented the main source of inspiration in drug development since ever. The review is well written and comprehensive, sometimes even too detailed, but presents significant flaws, two major and few minor, that have to be stressed.
1. Novelty. Alkaloids, starting from cited galantamine and reserpine, have always represented a resource for medicine first and now mainly a starting point for drug design. Therefore, in the literature there are plenty of research articles regarding alkaloids’ SAR as well as reviews summarizing them. Among these, a lot of reviews have been published in last years on alkaloids’ neuroprotective properties and some of them are cited in the manuscript but many are not:
- 2020. Current Medicinal Chemistry, 2020, 27, 5887-5917
- 2021. Bioorganic & Medicinal Chemistry 43 (2021) 116270
- 2021. Biomedicine & Pharmacotherapy 139 (2021) 111609
- 2021. Molecules 2021, 26, 728.
Herein there are reported only four examples, but there are even more, beside the comprehensive book chapter of 2018 (cited, https://doi.org/10.1016/B978-0-444-64183-0.00008-7). Not all are the same, some reviews are more general (i.e., natural products) and other more specific (i.e., spirocyclic, chemical-based) but in this reviewer’s opinion there is no need of another alkaloid’s review, even if this manuscript in some paragraph is more specific and detailed.
2. Nomenclature. The development of Multi-Target-Directed Ligands refers to the design strategy (or approach) where two (or more) pharmacophores are combined (linked, merged, fused, ….) with resulting polyhedral/synergistic biological outcome due to simultaneous interactions of pharmacophores with the “respective” targets. In parallel, there are privileged structures, such as usually happen for natural compounds, able per se to interact with multiple targets, therefore exhibiting a multitarget profile. In this case, there are no distinct pharmacophores but a same structure that can exert a pleiotropic activity. As a result, it is not correct to recall natural compounds as MTDLs, where multitarget compounds sounds more appropriate. Please modify accordingly along the manuscript.
3. The introduction section is too detailed and, in some part, heavy to read, because there are some details that are not essential for the reader which makes it difficult to read (e.g., section regarding different AChE enzymes). Besides the importance of introducing first all the involved potential targets, there is no need of 5 pages extensively describing etiopathology of AD. It is enough to present each interested target and their networks without describing in detail all the pathological cascades. The same happens in the rest of the manuscript where too details are likely to weaken reader’s attention and lose key points. It is better to highlight most important activities and only cite the other biological properties.
4. Please correct the reference section because they are numbered twice.
Based on these premises, especially the first point, I suggest to not accept this manuscript for publication on International Journal of Molecular Sciences.
Author Response

(The authors gave the same response as above.)

Reviewer 5 Report
A very good paper. A little verbose in the présentation of the pathology of AD, could have been a little more concise, nevertheless a very welcome refresher for readers less clear on the mechanism and pharmacology of AD. The pharmacognostic part however is a pleasure to read. Well documented and clear enough.
It could have benefited by a summarizing table and perhaps a visual aid like maybe a Venn diagram of overlapping mechanisms for various plant components, but it is just an opinion.
It can be published as it is. Very serious work.
Author Response

(The authors gave the same response as above.)

Round 2
Reviewer 4 Report
There are not any changes in the revised version that could change this reviewer's opinion. Furthermore, I agree that alkaloids can be defined multitarget agents, as indicated in previous revision, but MTDLs remain different, as described in previous revision.
Author Response
Dear Editor,
For the second time, we have carefully revised the manuscript entitled “Natural Alkaloids as Multi-Target-Directed Ligands Towards Factors Implicated in Alzheimer's disease“ Manuscript ID: ijms-2210183, according to the reviewers' suggestions. All additional changes are now highlighted in green and are visible in the MS Word’s “Track Changes” function.
Due to the reviewer's #3 strong opinion, we have replaced the term “Multi-Target-Directed ligands” throughout the whole article with the term “multi-target compounds”. This change was also incorporated into the title. Additionally, we had to slightly edit the abstract to incorporate and explain the difference between these terms.
We would like to keep the other parts of the manuscript as they are, given that the other four opponents had no comments on these parts. Thank you for your understanding.
In light of these additional changes, we are positive once again that our revised manuscript meets the criteria to be published in the International Journal of Molecular Sciences and would be of interest to all readers from the scientific community.
On behalf of all authors
Yours Sincerely,
Lucie Cahlíková, prof.

Round 3
Reviewer 4 Report
I appreciated finally the correction about the nomenclature that the authors made as requested, but, as I explained during the first revision, this manuscript completely misses the criteria of novelty that a paper published in high-impact journals like International Journal of Molecular Sciences should have.
Therefore, from my point of view there is no significant change in respect to my previous round of revisions.
Author Response
Dear Editor,
we are sending our comments on the review report of one of the five opponents of our revised manuscript entitled “Natural Alkaloids as Multi-Target Compounds Towards Factors Implicated in Alzheimer's disease“ Manuscript ID: ijms-2210183.
Reviewer's #3
I appreciated finally the correction about the nomenclature that the authors made as requested, but, as I explained during the first revision, this manuscript completely misses the criteria of novelty that a paper published in high-impact journals like International Journal of Molecular Sciences should have.
Therefore, from my point of view there is no significant change in respect to my previous round of revisions.
Dear reviewer,
we accept your point of view, we leave the final decision on the article to the editors. We believe that our article summarizes the latest knowledge about natural alkaloids as Multi-Target Compounds for AD and meets criteria to be published in IJMS. More than 37 percent of the cited references have been published in the last five years; our review article covers the latest results in the described area.
We are positive that our revised manuscript meets the criteria to be published in the International Journal of Molecular Sciences and would be of interest to all readers from the scientific community.
On behalf of all authors
Yours Sincerely,
Lucie Cahlíková, prof.
